



# On the zero-level offset in the GOSAT TANSO-FTS O₂ A-band and the quality of solar-induced chlorophyll fluorescence (SIF): Comparison of SIF between GOSAT and OCO-2

Haruki Oshio[1], Yukio Yoshida[1], and Tsuneo Matsunaga[1]

[1]Center for Global Environmental Research, National Institute for Environmental Studies, Tsukuba, 305-8506, Japan

*Correspondence to*: Haruki Oshio (oshio.haruki@nies.go.jp)

**Abstract.** Satellite remote sensing of solar-induced chlorophyll fluorescence (SIF) has attracted attention as a method for improving the estimation accuracy of the photosynthetic production of terrestrial vegetation in recent years. The Greenhouse
gases Observing SATellite (GOSAT) has the ability to observe both SIF and the concentrations of $CO_2$ and $CH_4$ and thus is expected to contribute to understanding of the global carbon budget. Evaluating artifact signals (e.g., zero-level offset caused by non-linearity in the analogue circuit in the case of GOSAT) is effective for inferring the instrument status and important for retrieving SIF from satellite measurements. Here we investigate the criteria for identifying vegetation-free areas to evaluate the zero-level offset and the offset correction method, while comparing the derived SIF with the Orbiting Carbon
Observatory-2 (OCO-2) SIF at multiple spatial scales (footprint to global). The criteria were determined as a small variation in the radiance within the GOSAT instantaneous field of view for cloudy ocean scenes and a higher albedo in the 2.0 μm band than in the 1.6 μm band for bare soil scenes, which were slightly different from the previously used criteria. The GOSAT SIF that was most consistent with the OCO-2 SIF was obtained when the zero-level offset was evaluated from bare soil over the globe, with a bias of about 0.1 mW m$^{-2}$ nm$^{-1}$ sr$^{-1}$. Our results agree with the previous comparisons and support
the consistency among the present satellite SIF data, which is important for the utilization of those data. An analysis of the temporal variation of the zero-level offset over a period of 9 years suggests that the radiometric sensitivity of the GOSAT spectrometer changed after switching the optics path selector in January 2015.

## 1 Introduction

Accurate estimation of the photosynthetic production of terrestrial vegetation is important to understand and predict the
global carbon cycle and climate change. Global observation of solar-induced chlorophyll fluorescence (SIF) from satellite data has attracted attention as a potential means of reducing the uncertainties in the estimation of photosynthetic production (Stoll et al., 1999; Moya et al., 2004). SIF is a weak radiation emitted by chlorophylls during the photosynthesis process and thus is considered to be a better proxy for photosynthetic activity than the conventional vegetation indices. Frankenberg et al.



(2011a) showed that the variation in fractional depth (filling-in) of the Fraunhofer line by SIF can be detected by satellite sensors having a high spectral resolution at the wavelength domain without the telluric absorption lines in the $O_2$ A-band. Since then, SIF retrievals have been performed by retrieving the signal that leads to filling-in of the Fraunhofer line (filling-in signal) from the spectra of the Thermal And Near infrared Sensor for carbon Observation-Fourier Transform Spectrometer

(TANSO-FTS) onboard the Greenhouse gases Observing SATellite (GOSAT) (Joiner et al., 2011; Frankenberg et al., 2011b). GOSAT was launched on January 23, 2009 and has been operating for more than ten years. The primary target of its measurement is the concentration of carbon dioxide and methane in the Earth's atmosphere.

Spectra of the TANSO-FTS $O_2$ A-band include the zero-level offset, which is a spectrally constant additive signal caused by non-linearity in the analogue circuit and the analog-to-digital converter (ADC) (Kuze et al., 2012). The intensity

of the zero-level offset varies according to the radiance input to the TANSO-FTS (Kuze et al., 2012). Frankenberg et al. (2011b) retrieved the filling-in signal from the spectra of the TANSO-FTS over Antarctica (SIF = 0) and showed that the zero-level offset is on the same level as SIF and thus should be corrected to obtain SIF. The zero-level offset is also a non-negligible factor for the retrieval of column-averaged dry-air mole fractions of carbon dioxide ($XCO_2$) and methane ($XCH_4$) from the multiple bands of the TANSO-FTS as it affects the retrieval of aerosol parameters and surface pressure from the $O_2$

A-band (Frankenberg et al., 2012). Moreover, the instrument status—e.g., the level of radiometric degradation—is reflected in the relationship between the zero-level offset and the observed radiance, which might have been affected by the pointing mechanism switch in January 2015 (Kuze et al., 2016). By using cloudy ocean data and bare soil data, different filling-in signals have been obtained between the northern and the southern hemispheres and between different months (Joiner et al., 2012; Guanter et al., 2012). However, studies that focus on the characteristics of the zero-level offset have been limited to

the initial period of the GOSAT observation. Furthermore, detailed evaluations have not been conducted on the criteria for identifying vegetation-free areas and the spatiotemporal variation of the filling-in signal.

To date, retrievals of SIF have been performed for other satellite sensors such as the Scanning Imaging Absorption spectroMeter for Atmospheric CHartographY (SCIAMACHY) (Joiner et al., 2012; Köhler et al., 2015), the Global Ozone Monitoring Instrument 2 (GOME-2) (Joiner et al., 2013; Joiner et al., 2016), the Orbiting Carbon Observatory-2 (OCO-2)

spectrometer (Frankenberg et al., 2014; Sun et al., 2018), the Atmospheric Carbon dioxide Grating Spectroradiometer (AGCS) onboard Chinese Carbon Dioxide Observation Satellite Mission (TanSat) (Du et al., 2018), and the TROPOspheric Monitoring Instrument (TROPOMI) (Köhler et al., 2018a). These retrievals have revealed the potential advantages of SIF for use in GPP estimation—namely, SIF has a strong linear relationship with the flux tower-derived GPP (Guanter et al., 2014; Sun et al., 2017; Li et al., 2018b; Zhang et al., 2018), the terrestrial biosphere models can be improved by assimilating SIF

(Parazoo et al., 2014; Zhang et al., 2014; Thum et al., 2017; MacBean et al., 2018), and SIF is related to the response of vegetation to drought stress  (Lee et al., 2013; Yoshida et al., 2015; Sun et al., 2015; Li et al., 2018a). The spatial and temporal resolution, observation time, viewing direction, and observed wavelength and its resolution differ among satellite sensors. Evaluating the quality of SIF data and the consistency among SIFs obtained by different sensors is important to fully utilize those data to elucidate the photosynthesis activity of the terrestrial vegetation. In the present study, we focus on an



inter-sensor comparison of SIF data, since no reliable methods for validating satellite-derived SIF using ground observations have been developed. GOSAT SIF is considered to be a more independent target for comparison, because its spectrum is obtained by FTS while other sensors are based on diffraction grating. Although agreement among satellite-derived SIF data has already been partly confirmed (Joiner et al., 2012; Joiner et al., 2013; Köhler et al., 2015; Köhler et al., 2018b; Du et al.,

2018), there were limitations in the simultaneity of these observations, the overlap of the footprints, and the global coverage. These limitations were significantly reduced in a recent comparison between TROPOMI SIF and OCO-2 SIF (Köhler et al., 2018a).

This study sought to evaluate the zero-level offset in the GOSAT TANSO-FTS $O_2$ A-band from the initial period of the GOSAT observation until recently and to investigate the consistency between the GOSAT-derived SIF and the SIF

derived from other satellite sensors. For the filling-in signal, the spatiotemporal variation and the difference among spectra identified by various criteria for vegetation-free areas are examined, and based on the results, several kinds of zero-level offset correction are conducted (Sect. 3). By comparing the derived SIF with the OCO-2 SIF, an appropriate correction method for the zero-level offset and the consistency of SIF are investigated (Sect. 4). OCO-2 is chosen for the comparisons because it is expected to have small random error due to the large amount of data. Moreover, OCO-2 SIF has previously

been compared with airborne data (Sun et al., 2017), GOME-2 (Köhler et al., 2018b), TanSat, (Du et al., 2018), and TROPOMI (Köhler et al., 2018a). Comparisons between GOSAT SIF and OCO-2 SIF are performed at multiple spatial scales (footprint to global) while considering the difference in observation patterns and the characteristics of vegetation.

## 2 Materials and data processing

### 2.1 GOSAT

GOSAT was launched on 23 January 2009 and is on a sun-synchronous orbit at 666-km altitude with 3-day recurrence and a descending node around 13:00 local time. It is equipped with two instruments: a Fourier transform spectrometer (TANSO-FTS) and a Cloud and Aerosol Imager (TANSO-CAI). The TANSO-FTS has three bands in the Short-Wavelength InfraRed (SWIR) region (an $O_2$ A-band, a weak $CO_2$ absorption band, and a strong $CO_2$ absorption band (Bands 1, 2, and 3) centred at 0.76, 1.6, and 2.0 μm, respectively) and records two orthogonal polarization components (hereafter called P/S components).

The spectral resolution (sampling interval) of Bands 1, 2, and 3 are about 0.011 to 0.012 nm, 0.045 to 0.064 nm, and 0.069 to 0.090 nm, respectively. The full widths at half-maximum (FWHMs) of the instrument line shape function (ILSF) of Bands 1, 2, and 3 are about 0.02 nm, 0.06 nm, and 0.10 nm, respectively. For the signal processing of the TANSO-FTS, the amplifier gain level can be controlled at three different levels: high (H), medium (M), and low (L), according to the brightness of the target. The instantaneous field of view (IFOV) of the TANSO-FTS is 15.8 mrad, which corresponds to a circular surface

footprint of about 10.5 km in diameter at nadir. The TANSO-FTS has a pointing mechanism with mirror driving angles of ±35° in the cross-track direction and ±20° in the along-track direction. The optics path selector was changed from the primary one to the secondary one on 26 January 2015 as the along-track pointing of the primary system worsened. The



TANSO-CAI is a push-broom imager and has four narrow bands in the near-ultraviolet to near-infrared regions centred at 0.38, 0.674, 0.87, and 1.6 μm with spatial resolutions of 0.5, 0.5, 0.5, and 1.5 km, respectively, for nadir pixels.

The TANSO-FTS L1B product (radiance spectral data) version V201.202 was used in the present study. We corrected the sensitivity degradation of the TANSO-FTS using a radiometric degradation model that is based on the on-orbit

solar calibration data (Yoshida et al., 2012). The retrieval window was 756.0 to 759.1 nm (the short wavelength side of band 1), within which there are four strong Fraunhofer lines without telluric absorption lines, which is a strategy similar to that of Frankenberg et al. (2011a). In this wavelength domain, the sampling interval is about 0.011 nm, and the FWHM of the ILSF is about 0.02 nm. The retrieval window around 771 nm, where weak $O_2$ absorption lines are located, was not used in the present study. The filling-in signal was retrieved in a similar manner as in previous studies that fitted the modelled and the

observed spectra using a simple forward model (Frankenberg et al., 2011a; Joiner et al., 2011). The retrieval algorithm is based on a non-linear maximum a posteriori estimation, just as for the TANSO-FTS L2 SWIR $XCO_2$ and $XCH_4$ retrievals (Yoshida et al., 2011; Yoshida et al., 2013; Yoshida et al., 2017). The state vector consisted of a spectrally constant filling-in signal, spectrally linear surface albedo, and wavenumber dispersion. A clear sky was assumed in the forward model. Retrieval was conducted separately for the P- and S-polarization spectra, although a scalar radiative transfer code was used

as the forward model. Precision error in each retrieved filling-in signal was calculated by error propagation assuming only the instrumental random noise. When calculating the mean value, the standard error was also calculated by error propagation by assuming the independence of each error.

The TANSO-CAI L1B product (radiance data) and L2 product (cloud coverage data) were used to confirm the cloud condition. The cloud fraction within the TANSO-FTS IFOV was calculated based on the integrated clear confidence

level stored in the TANSO-CAI L2 product. The integrated clear confidence level is represented by a real number between 0 to 1 and given for each pixel of the TANSO-CAI. Pixels with an integrated clear confidence level lower than 0.33 were classified as cloudy pixels. The fraction of the cloudy pixels within the TANSO-FTS IFOV was then calculated.

Data satisfying all the following criteria were used for the subsequent analysis: (1) several data-quality flags stored in the L1B product and spectrum quality check utilizing the out-of-band spectra (Yoshida et al., 2017) are set as OK; (2) the

25 solar zenith angle (SZA) is < 70°; (3) the signal-to-noise ratio (SNR) of band 1 P- or S-polarization is ≥ 70; (4) the mean squared value of the residual spectrum is ≤ 2; (5) the land fraction of the TANSO-FTS footprint is 100% or 0%; and (6) the data are acquired with gain-H. Concerning the cloud screening, different criteria were used for evaluating the filling-in signal (Sect. 3) and comparison with OCO-2 (Sect. 4).

## 2.2 OCO-2

OCO-2 was launched on 2 July 2014 and is on a sun-synchronous orbit with 16-day recurrence and an ascending node around 13:30 local time. Its instruments consist of three grating spectrometers that measure spectra at the $O_2$ A-band, the weak $CO_2$ absorption band, and the strong $CO_2$ absorption band with sampling intervals of 0.015 nm, 0.031 nm, and 0.04 nm, respectively, and FWHM of 0.04 nm, 0.08 nm, and 0.10 nm, respectively. Each grating spectrometer has eight adjacent



footprints across the swath with each surface footprint of 1.29 km × 2.25 km at nadir. There are three observation modes: the nadir, glint, and target modes. The SIF lite product version B8100r was used in the present study. This product includes SIF retrieved separately from windows around 757 nm and around 771 nm. We used SIF from the former window, which is similar to the window for the SIF used by GOSAT. Only data with a cloud flag from the Oxygen A-Band Cloud Screening

Algorithm set to clear sky were used. The nadir mode and glint mode were used separately, since the SIF has been reported to be dependent on the viewing angle (Guanter et al., 2012; Köhler et al., 2018b; Zhang et al., 2018).

The retrieval window of OCO-2 SIF at 757 nm is 758.1 to 759.2 nm, which is not identical to that used by GOSAT, because the edge of the short wavelength side of the $O_2$ A-band is different between GOSAT and OCO-2. Therefore, it is expected that the GOSAT SIF is higher than the OCO-2 SIF because of the spectral shape of SIF. This difference was

neglected when comparing GOSAT SIF and OCO-2 SIF, since it was expected to be only 3% to 4% according to the observations at leaf level (Rascher et al., 2009; Magney et al., 2017) and simulations at the canopy level (Joiner et al., 2013; Verrelst et al., 2016). Although the equator crossing time is not very different between GOSAT and OCO-2, the difference in observation time becomes large for high latitude areas. The differences in the location and size of the footprint, observation time, and viewing angle between the two satellites were taken into account for the comparison. The detailed comparison

method is described in Sect. 4.1.

### 2.3 NDVI, LAI, and land cover type

We used the normalized difference vegetation index (NDVI) to identify vegetation-free areas for the zero-level offset correction (Sect. 3) and the leaf area index (LAI) and land cover type to select SIF data of GOSAT and OCO-2 suitable for comparisons (Sect. 4). The MODerate resolution Imaging Spectroradiometer (MODIS) products were used: NDVI monthly

L3 global 1 km product (MYD13A3; Didan, 2015), LAI 4-day L4 global 500 m product (MCD15A3H; Myneni et al., 2015), and land cover type yearly L3 Global 500 m product (MCD12Q1; Friedl and Sulla-Menashe, 2015). The land cover type derived by the International Geosphere-Biosphere Programme (IGBP) classification scheme was used. These data were transformed to a geographic grid of latitude 0.02° × longitude 0.02° by nearest neighbour resampling. For LAI, the dataset for each 4-day interval was adopted to calculate the monthly mean value if the 4 days were included in the month. NDVI and

LAI within the TANSO-FTS footprint was obtained by calculating the mean values of 5 × 5 pixels with the centre pixel including the centre of the TANSO-FTS footprint. The land cover type of the TANSO-FTS footprint was also defined using 5 × 5 pixels, but the detailed treatment differed between comparison methods (Sect. 4.1). For OCO-2, the LAI and land cover type were obtained from the single pixel value of each MODIS product, including the centre of the OCO-2 footprint.



## 3 GOSAT filling-in signal and zero-level offset correction

### 3.1 Current state of evaluation of the zero-level offset

The retrieved filling-in signal from the GOSAT spectra consists of SIF and the zero-level offset, and thus correction for the zero-level offset is required to derive SIF ($SIF = Filling\text{-}in\ signal - Zero\ level\ offset$, Frankenberg et al., 2011b). The

zero-level offset can be represented by the filling-in signal over vegetation-free areas ($SIF = 0$). Initially, the filling-in signal over Antarctica was used to evaluate the zero-level offset (Frankenberg et al., 2011b). The results of this analysis showed a strong correlation between the zero-level offset and the averaged radiance of the TANSO-FTS band 1. However, the spatiotemporal coverage of the Antarctica data was limited. A cloudy ocean area (Joiner et al., 2012) and bare soil area (Guanter et al., 2012) were subsequently used to account for the spatiotemporal variation of the zero-level offset. Together,

these two studies reported the characteristics of the zero-level offset, i.e., the difference between the northern and the southern hemispheres and the temporal variation (from the boreal summer of 2009 to the boreal spring of 2011). The TANSO-FTS L1B products version V050.050 to V110.110 were used in the above-mentioned studies. The newer version (V201.202) was used in the present study. Although the band 1 nonlinearity correction in the TANSO-FTS L1B product has been updated (Kuze et al., 2012; Kuze et al., 2016), the filling-in signal over Antarctica retrieved in the present study (Sect.

2.1) shows dependence of the zero-level offset on the observed radiance (Fig. S1), indicating that the zero-level offset correction is still required. In this study, the filling-in signal was investigated over areas identified by various criteria over a long period of time, to examine whether the criteria for identifying the cloudy ocean and bare soil areas used in the previous studies are appropriate and to evaluate the spatiotemporal variation of the zero-level offset.

### 3.2 Filling-in signal over cloudy ocean

To evaluate the zero-level offset from ocean data, Joiner et al. (2012) used data with a TANSO-CAI cloud fraction > 0.2 to reduce the ocean-derived filling-in (ocean Raman scattering or fluorescence). Here, we investigated the variation of the filling-in signal over ocean according to the cloud fraction. Figure 1 shows the monthly variation of the filling-in signal over ocean within 30°N to 45°N for a representative level of the averaged radiance of the TANSO-FTS band 1 (0.0005 to 0.0006 V (cm$^{-1}$)$^{-1}$ for P-polarization and 0.0006 to 0.0007 V (cm$^{-1}$)$^{-1}$ for S-polarization). The distribution of the band 1 averaged

radiance is presented in the supplement (Fig. S2). The filling-in signal varies according to the band 1 averaged radiance (Fig. S1), and the latitudinal variation is expected (Joiner et al., 2012); however, the variation pattern of the filling-in signal according to time and the cloud fraction was similar between levels of the band 1 averaged radiance and between latitude zones. For Fig. 1, the standard error of the monthly mean filling-in signal was about 0.5 to $1.5 \times 10^{-10}$ and 1.5 to $3.0 \times 10^{-10}$ W cm$^{-2}$ (cm$^{-1}$)$^{-1}$ sr$^{-1}$ for cloud fraction higher and lower than 0.75, respectively. If the influence of the ocean-derived filling-in

was dominant, an increase in the retrieved filling-in signal would be expected according to the decrease in cloud fraction. However, for both P- and S-polarization, the data using cloud fractions of 0.25 to 0.75 showed a significantly lower filling-in signal compared to the data using a cloud fraction of 1.0. This variation is larger than that expected to be caused by the





atmospheric Raman scattering (Joiner et al., 2012). A filling-in signal of $10^{-9}$ W cm$^{-2}$ (cm$^{-1}$)$^{-1}$ sr$^{-1}$ corresponds to 0.17 in SIF units (mW m$^{-2}$ nm$^{-1}$ sr$^{-1}$), meaning that the variation of the filling-in signal according to the cloud fraction corresponds to a variation in SIF of 0.3 to 0.5 mW m$^{-2}$ nm$^{-1}$ sr$^{-1}$. This variation is relatively large compared to the typical range of SIF over the globe (0 to 2 mW m$^{-2}$ nm$^{-1}$ sr$^{-1}$).

One possible reason for the variation of the filling-in signal according to the cloud fraction is the image motion compensation (IMC) fluctuation effect (Aoki et al., 2008). IMC helps the TANSO-FTS to look at the same position during the interferogram acquisition. However, the IMC mirror could vibrate, resulting in fluctuation of the radiance input to the TANSO-FTS if the albedo within and around the IFOV is nonuniform. This effect might cause discrepancy between the actual radiance input to the TANSO-FTS and the observed radiance when the IFOV includes cloud and ocean, yielding

different retrieved filling-in signals against the same observed radiance level. In contrast, cloudy ocean with a small variation in the TANSO-CAI radiance (cloud fraction > 0 and $VAR_{CAI} \leq 0.1$ in Fig. 1) offered a filling-in signal with small variation. $VAR_{CAI}$ is a ratio of the standard deviation to the mean value of the TANSO-CAI radiance within the TANSO-FTS IFOV. We suggest new criteria: cloud fraction > 0 and $VAR_{CAI} \leq 0.1$. The filling-in signal over ocean identified by these criteria was used as the zero-level offset evaluated from cloudy ocean in this study. This criteria is suitable from the point of view of

spatiotemporal coverage, since the data for cloud fraction 1.0 may be unavailable depending on season and latitude.

### 3.3 Filling-in signal over bare soil

To identify spectra over bare soil, Guanter et al. (2012) used criteria of $R_1 < R_2$ or $R_2 < R_3$, where $R_i$ is the top-of-atmosphere reflectance of the TANSO-FTS SWIR band $i$. We investigated the effectiveness of these criteria by using the filling-in signal over areas having different NDVI values. In the present study, the retrieved albedo values were used (hereafter $R_i$ denotes the

20 retrieved albedo value of band $i$). The filling-in signal varied according to the cloud fraction (Sect. 3.2), and thus only data with the TANSO-CAI cloud fraction of 0 were used in this section. Figure 2 shows the monthly variation of the filling-in signal derived from data identified by the different criteria of the retrieved albedo value and NDVI value. The variation of the filling-in signal according to the criteria of albedo values and NDVI values was similar between P- and S-polarization. The filling-in signal clearly increased with the increase in NDVI, showing the influence of SIF from terrestrial vegetation on

the filling-in. The filling-in signal for $R_1 < R_2$ was higher than that for $R_2 < R_3$ or low NDVI values. It seems that SIF was included in the filling-in signal for $R_1 < R_2$. These criteria identify a large number of data but are considered to identify vegetated areas (Fig. 3(a)). The filling-in signal for $R_2 < R_3$ was comparable to that for low NDVI. According to Fig. 3(b), the data identified by criteria $R_2 < R_3$ were limited to specific areas that appeared to be barren. These results indicate that the criteria of $R_2 < R_3$ offer a small number of data but are robust for the identification of vegetation-free areas.

To expand the spatial coverage and the number of data, criteria based on the NDVI value were investigated. The difference in the monthly mean filling-in signal was calculated between the data identified by the criteria based on the NDVI value and the data identified by the criteria of $R_2 < R_3$, and the difference was averaged over a target time period (May 2009 to January 2018) (Fig. 4). The calculation was conducted for each observed level of TANSO-FTS radiance (horizontal axis



of Fig. 4). The difference was almost 0 for NDVI ≤ 0.2, irrespective of the observed TANSO-FTS radiance and polarization. The criteria of NDVI ≤ 0.2 offered a larger number of data and wider spatial coverage compared to the criteria of $R_2 < R_3$ (Fig. 3(b)(c)). The difference shown in Fig. 4 became larger for NDVI > 0.2, especially in the case of a high radiance level. For NDVI 0.2 to 0.3, the difference was comparable to that for the criteria of $R_1 < R_2$ (black dotted line in Fig. 4). For these

cases, the difference was within 0.1 in SIF units (mW m$^{-2}$ nm$^{-1}$ sr$^{-1}$). It should be noted that the difference shown in Fig. 4 is the time mean value, and this difference became large in summer (Fig. 2). Moreover, another set of criteria using NDVI were tested: NDVI$_{max}$ ≤ 0.3 for 5 ×5 pixels around the centre of the TANSO-FTS footprint. The filling-in signal for these criteria showed a similar tendency to that for mean NDVI ≤ 0.2 (gray solid line in Fig. 4), and these criteria offered a larger number of data (Fig. 3(d)). We suggest a new set of criteria: $R_2 < R_3$ or mean NDVI ≤ 0.2 or NDVI$_{max}$ ≤ 0.3. The filling-in signal

identified by these criteria was used as the zero-level offset evaluated from bare soil in this study. We accepted data with a cloud fraction ≤ 0.2, because the results hardly varied within this cloud fraction (Fig. S3 and Fig. S4).

### 3.4 Spatiotemporal variation of zero-level offset

The zero-level offset evaluated from cloudy ocean (Fig. 1) and bare soil (Fig. 2) for P-polarization showed a sharp decrease after GOSAT started its observations. In contrast, the variation in S-polarization was small. Frankenberg et al. (2011a) used

only S-polarization data because the zero-level offset for P-polarization showed significant time-dependence. On the other hand, Guanter et al. (2012) reported a decrease in the zero-level offset for both P- and S-polarization. Our results agree with those of Frankenberg et al. (2011a). The decrease in P-polarization became gentle at around November 2011. For both polarizations, the gradient of the temporal variation of the zero-level offset changed in February 2015 following the switch from a primary to a secondary TANSO-FTS optics path selector on 26 January 2015.

A possible cause of such temporal variation is a variation in the instrumental characteristics related to the zero-level offset itself (analogue circuit and ADC) and to the radiance (radiometric degradation of the TANSO-FTS). The degradation does not directly change the fractional depth of the Fraunhofer line; however, a discrepancy between the actual radiance input to the TANSO-FTS and the observed radiance affects the relationship between the zero-level offset value and the observed radiance. The initial decrease in P-polarization appears to mainly relate to the variation in characteristics of the

analogue circuit and the ADC. This is supported by the fact that the radiometric degradation model was generated using the on-orbit solar calibration data covering the time period when the zero-level offset was decreasing (Yoshida et al., 2012). In addition, the initial decrease was observed only for P-polarization, although the degradation correction was conducted in the same way for both polarizations. Of course, there is a possibility that the discrepancy between the degradation model and the actual degradation is not minimal, since it is difficult to accurately model the rapid radiometric degradation using the

monthly on-orbit calibration data (Yoshida et al., 2012). There is also a possibility that the accuracy of the degradation model differs between P- and S-polarization. On the other hand, the decrease in the zero-level offset from February 2015 for both polarizations seems to have been caused by radiometric degradation of the TANSO-FTS. The degradation correction





factors are predicted values after about 3 years from the launch of GOSAT under the assumption that the degradation becomes slower with time in general. It is necessary to reevaluate the degradation after the optics path selector is changed.

In addition to the aforementioned general variation, periodic variation peaking at around June was observed for the zero-level offset evaluated from cloudy ocean (Figs. 1 and S5; the right panel of Fig. S5 shows the extraction of the zero-level offset (red square in Fig. 1)). The periodicity identified by each set of criteria is not clear for the bare soil data (Fig. 2), but it became clearer when aggregating the results from each set of criteria with offset values slightly different from those for cloudy ocean (Fig. S5). Figure 5 shows the latitudinal variation of the monthly mean zero-level offset. For cloudy ocean, the offset value increased toward the low latitude area. No clear latitudinal dependence was observed for bare soil, although the narrower coverage of the latitudinal zone and the larger standard error than for cloudy ocean might have obscured the dependence.

The periodic variation (Guanter et al., 2012) and difference between the hemispheres (Joiner et al., 2012) were reported in previous studies. As discussed in those studies, the cause of such variation in the zero-level offset seems to be the instrumental effects that vary with orbit phase and season (e.g., the detector temperature and condition of analogue circuits) and rotational-Raman scattering (RRS). We investigated the variation of the zero-level offset against the detector temperature and found no clear relationship (Fig. S6 and Fig. S7). A large sampling step of the recorded temperature (~0.08 K) might obscure the relationship. The aforementioned periodic and latitudinal variation correspond to the increase in the zero-level offset according to the decrease in SZA, which is the opposite of the tendency for the expected filling-in by RRS. Therefore, the difference in the zero-level offset between cloudy ocean and bare soil cannot be simply explained by the difference in RRS between ocean and land. Moreover, assumptions regarding the retrieval, solar irradiance model and unpolarized forward model could affect the retrieved filling-in and albedo, respectively, but do not seem to account for the spatiotemporal variation of zero-level offset. We therefore decided to use the zero-level offset evaluated from cloudy ocean and that from bare soil without corrections for their spatiotemporal variation. The differences in latitudinal dependence (Fig. 5) and absolute value (Fig. S5) between cloudy ocean and bare soil will be further investigated in a future study.

## 3.5 Calculating SIF by correcting the zero-level offset

For each TANSO-FTS spectra, SIF was calculated separately for P- and S-polarization by subtracting the zero-level offset from the filling-in signal. The zero-level offset was determined using a table representing the variation of the zero-level offset according to time and the averaged radiance of the TANSO-FTS band 1. The table was generated separately for P- and S-polarization by binning the zero-level offset against the averaged radiance with an interval of 0.00003 V $(cm^{-1})^{-1}$ for each month. The mean value and standard error were calculated for each bin. The binned offset values were used without smoothing to take into account the aforementioned temporal variation. SIF calculation was conducted unless the offset value of the bin was derived from less than 10 data points. Subsequently, the SIF derived from P- and S-polarization were averaged under the assumption that the difference in SIF between the polarizations is minimal.



Four different offset tables were tested: (1) a table generated using the zero-level offset evaluated from Antarctica, Greenland, cloudy ocean, and bare soil, (2) a table similar to (1) but generated separately for each latitude zone (15° interval), (3) a table generated using the zero-level offset evaluated from bare soil, and (4) a table similar to (3) but generated separately for the northern and the southern hemisphere (hereafter called Corrections 1, 2, 3, and 4, respectively). Correction 2 is based on the latitudinal dependence of the zero-level offset (Fig. 5(a)) and is a similar strategy to that used by Joiner et al. (2012). Correction 3 is based on the difference in zero-level offset between cloudy ocean and bare soil (Fig. 5 and Fig. S5). Although latitudinal dependence was not clear for bare soil (Fig. 5(b)), we tested Correction 4 to investigate whether the latitudinal dependence is dominant and should be taken into account.

## 4 Comparison between GOSAT SIF and OCO-2 SIF

### 4.1 Comparison method

The SIF values derived from satellite observation vary according to the observation time, viewing direction, observed wavelength, and type and condition of vegetation. When comparing SIF data among satellite sensors, it is necessary to account for each of these factors. Joiner et al. (2012) compared the global distribution and seasonal cycle for specific regions between GOSAT SIF and SCIAMACHY SIF while considering the difference in wavelength. In a subsequent study, the same authors (Joiner et al. 2013) compared the global distribution of SIF derived from GOME-2 with that from GOSAT. Recently, more refined comparisons have been performed. Although their analysis was limited to the Amazon region, Köhler et al. (2018b) compared OCO-2 SIF and GOME-2 SIF using data with a footprint covering the target land cover type (evergreen broadleaf) and considering the difference in observation geometry (the angle of incident sunlight and observation). Köhler et al. (2018a) also compared TROPOMI SIF with OCO-2 SIF using matching criteria in which the OCO-2 footprint was included in the TROPOMI footprint and the difference in overpass time and observation geometry was small [10 min and 20° in phase angle (the angle between the observation direction and the incident sunlight), respectively]. Normalization of SIF is effective to account for the difference in observation time between satellites. A simple approach is to divide SIF by cos(SZA) (as in Joiner et al., 2011). Frankenberg et al. (2011b) proposed an approximation formula to obtain the daily average SIF from the observed instantaneous SIF and the variation in SZA throughout a day. Both normalizations were based on the same principle to account for the variation in incident photosynthetically active radiation (PAR) under the assumption of a clear sky. Dividing by cos(SZA) apparently amplified the noise for SIF acquired under large SZA. The daily average SIF seems to be suitable when comparing with GPP.

Based on the aforementioned studies, we conducted three different comparisons between GOSAT and OCO-2: in Case 1 used a footprint level with strict matching criteria, Case 2 used a specific region with similar observation time, and Case 3 had a global scale (Sect. 4.2, 4.3, and 4.4, respectively). In Case 1 a simple comparison was performed using overlapped data. The matching criteria were that the OCO-2 footprint was included in the GOSAT footprint (the distance was less than 5 km between the footprint centre of the satellites) and the difference in observation time was within 15 min.



OCO-2 SIFs satisfying these matching criteria were identified and averaged. To minimize the error due to the heterogeneity of the land cover, the land cover types within the satellite footprints were checked. If the majority of the land cover types were the same and the composition ratio of the land cover types was similar between GOSAT and OCO-2, the SIFs were used for comparison. The majority and composition ratio were defined against $5 \times 5$ MODIS pixels for GOSAT and the identified footprints for OCO-2. The criterion for the composition ratio was that the difference in the fraction of land cover type between GOSAT and OCO-2 was less than 0.2 for each IGBP type. The OCO-2 SIF derived from the nadir mode was used to minimize the influence of the difference in observation geometry. It is difficult to investigate the influence of observation geometry on SIF by comparing GOSAT SIF, OCO-2 nadir SIF, and glint SIF, because the number of data are expected to be small for the strict criteria of Case 1.

The land cover type and the number of data are limited in Case 1, because the GOSAT path and OCO-2 path cross within a specific latitude zone (around 30°N). Case 2 made comparisons for specific regions having different vegetation coverage where the difference in the observation time was small. The target regions and land cover types were evergreen broadleaf forest in Southeast Asia, cropland in the Corn Belt region of the USA, and grassland in the southern part of the USA. GOSAT SIF data included in the target region and having a fraction of the target land cover type of $\geq 0.8$ within the footprint were identified and averaged for each month. OCO-2 SIF data included in the target region and having the target land cover type were identified and averaged for each month. OCO-2 SIFs derived from the nadir mode and the glint mode were averaged separately. The mean LAI and the mean local time of the observation were calculated in the same way as SIF.

Case 3 made comparisons over the globe to confirm that the spatial variation of the zero-level offset was effectively corrected. The monthly mean SIF was calculated for each land cover type within a grid box of 5° latitude $\times$ 10° longitude. Concerning GOSAT, only SIF data for which the fraction of the land cover type was $\geq 0.8$ within the footprint were used. For each grid, the land cover type containing the maximum number of GOSAT data was defined as the representative land cover type. SIF of the representative land cover type was compared between GOSAT and OCO-2. The mean LAI and the mean local time of the observation were calculated in the same way as SIF. If the difference in LAI between GOSAT and OCO-2 exceeded 20%, the grid was not used for comparison. For this comparison, OCO-2 SIF derived from the nadir mode was used.

For each comparison, the phase angle was calculated to confirm the effect of observation geometry, as in Köhler et al. (2018b). Normalization that divides SIF by cos(SZA) was applied. We conducted a simple normalization, because the present study sought only to evaluate the difference in instantaneous SIF between satellite data. Only GOSAT data with a cloud fraction $\leq 0.2$ were used. Comparisons were conducted for February 2015 to January 2018, because the GOSAT optics path selector was changed from the primary one to the secondary one in January 2015, and OCO-2 data were available from September 2014.



## 4.2 Case 1: Comparison at the footprint level

Figure 6 shows a scatter plot of GOSAT SIF and OCO-2 SIF. The GOSAT SIF derived by Correction 1 (Sect. 3.5) is depicted because its plot exhibited almost no variance under the different correction methods. The standard error of OCO-2 SIF differed among the data according to the number of footprints used. The precision error of the single GOSAT SIF was

0.5 to 0.7 mW m$^{-2}$ nm$^{-1}$ sr$^{-1}$, making the plot dispersive; however, deviation of the plot according to the land cover type was not found. When GOSAT SIF data having the absolute value of the matched OCO-2 SIF $\leq 0.1$ mW m$^{-2}$ nm$^{-1}$ sr$^{-1}$ were identified and averaged, the standard error was 0.077 mW m$^{-2}$ nm$^{-1}$ sr$^{-1}$ (the number of data points was 68). The standard error used in the present study is the estimated value from the instrumental random noise. The standard error calculated using the actual variation (standard deviation of the identified GOSAT SIF divided by the square root of the number of data) was

0.097 mW m$^{-2}$ nm$^{-1}$ sr$^{-1}$. The actual variation was slightly larger than the estimated value, since the estimation was based on only instrumental random noise. However, the estimated standard error appeared to explain the primary variation. The statistical analysis of the relationship between GOSAT SIF and OCO-2 SIF is shown in Table 1. The difference between the offset correction methods was small. Although the correlation between GOSAT SIF and OCO-2 SIF was weak, the mean difference was close to 0. The number of data points for Correction 4 was small, because the coverage of the radiance level

was narrow.

        Next, we calculated the monthly mean of SIF and the monthly mean of the difference between GOSAT SIF and OCO-2 SIF to investigate the temporal pattern (Fig. 7). The phase angle and latitude of the centre of the GOSAT footprint were also calculated. Averaging was conducted without taking the year into account. Figure 7(b) shows that the difference varied around zero with no clear cycle or seasonal pattern. Although the difference between the zero-level offset correction

methods was small, the difference value derived by Correction 1 deviated slightly from the others. It appeared that the zero-level offset evaluated from various targets over the globe differed from the actual zero-level offset in some cases. The difference between GOSAT SIF and OCO-2 SIF became large in November and December. This difference was larger than the standard error and the typical range of SIF over the globe (0 to 2 mW m$^{-2}$ nm$^{-1}$ sr$^{-1}$). According to Fig. 7(a), GOSAT SIF changed significantly between November and December, although OCO-2 SIF was almost stable. We checked the individual

GOSAT data (location, land cover, topography, observation angle, difference in SIF between P- and S-polarization, and so on) but found no clue to explain the large deviation. This trend was hardly changed by changing the matching criteria between the GOSAT data and OCO-2 data (the landcover, topography, observation angle, and so on). Köhler et al. (2018b) calculated variations in SIF according to the phase angle using a three-dimensional radiative transfer model in the Amazon. Their results showed that SIF decreased significantly when the phase angle changed from 0° (hot spot) to around 20°, and

then gently decreased when the phase angle became larger. Therefore, the effect of observation geometry on the results of this section seems to be small, because the phase angle was larger than 20° for most cases and the difference in the phase angle between GOSAT and OCO-2 was about 10° (Fig. 7(c)). Moreover, most of the SIF data used in this section appeared



to be derived from land cover types with a simple structure (Fig. 6), and hence the effect of observation geometry was negligible, as shown in a later section comparing the OCO-2 nadir and glint SIF.

The above results confirm that GOSAT SIF and OCO-2 SIF agree on the zero level when overlapped footprints with similar observation times are used. The latitude zone was limited to 30°N to 40°N and the land cover mainly consisted of barren areas in this section. The next section considers comparisons made using vegetated areas where the difference in observation time was small to investigate whether the agreement between GOSAT SIF and OCO-2 SIF is observed under high SIF emission, and whether the large difference found in November and December arise for other targets.

### 4.3 Case 2: Comparison for specific regions with similar observation times

Figure 8 shows the monthly variation of GOSAT SIF and OCO-2 SIF for three different vegetated areas. Panel (a) shows the variation for the evergreen broadleaf forest in Southeast Asia. In this region, the observation time of GOSAT was slightly earlier than that of OCO-2, but the difference was small (about 15 to 25 min). Although the difference in LAI was small between OCO-2 nadir and glint, the nadir SIF was higher than the glint SIF, especially in summer, revealing the effect of observation geometry (more shaded leaves were seen from glint observation than from nadir observation). Although fluctuations were found in GOSAT SIF, both GOSAT SIF and OCO-2 SIF showed a clear seasonal cycle, and the GOSAT SIF value was on a similar level as the OCO-2 nadir. When comparing the GOSAT and OCO-2 nadir SIFs, the effect of observation geometry seems to have been small, considering the phase angle. LAI was slightly higher for GOSAT than for OCO-2; however, the influence of this difference in LAI on SIF appeared to be small for such a high LAI condition (Koffi et al., 2015; Verrelst et al., 2016; Liu et al., 2017). The difference between the GOSAT SIF and OCO-2 nadir SIF did not correspond to the difference in LAI. The large difference in SIF in winter found in Sect. 4.2 was not observed. However, the GOSAT SIF was significantly higher than the OCO-2 nadir SIF for two months in summer (June 2016 and July 2017). This finding appeared to be attributable to errors in the zero-level offset correction or random error, since the effects of other factors seemed to be small, as discussed above. The SIF derived from Correction 4 showed large fluctuation, because the number of data for the offset table appeared to be insufficient.

The results for cropland in the USA are shown in Fig. 8(b). In this region, the observation time of GOSAT was slightly later than that of OCO-2, but the difference was slight. The difference in LAI and that in SIF between OCO-2 nadir and glint were also slight, indicating that the effect of observation geometry was small. In addition, the phase angle was larger than 20°, and thus the effect of observation geometry was minimal when comparing GOSAT and OCO-2 in this region. The seasonal cycle of GOSAT SIF corresponded to that of OCO-2. The GOSAT SIF values of Corrections 1 and 2 were very close to OCO-2 SIF. Corrections 3 and 4 offered lower SIF values compared to OCO-2. The variation of GOSAT SIF of the cropland was smoother compared to that of the evergreen broadleaf forest in Southeast Asia, although the standard errors were comparable.

Figure 8(c) shows the comparison for grassland in the USA. The observation time was almost the same between GOSAT and OCO-2 in this region. OCO-2 glint SIF was higher than the nadir SIF for 2015 and 2016. This difference seems





to correspond to the difference in LAI (e.g., there was a clear correspondence in May 2015 and the spring of 2016). The effect of observation geometry was expected to be minimal for the grassland, as discussed for the USA cropland. GOSAT SIF agreed with OCO-2 glint SIF both with respect to the seasonal cycle and the absolute value. The slightly higher SIF for GOSAT seemed to be explained by the difference in LAI. The correspondence between the variation in LAI and that in SIF

supports the consistency among the GOSAT SIF, OCO-2 nadir SIF, and glint SIF. Concerning the finding that the GOSAT SIF value was lower than 0 in December 2016, this was probably attributable to the random error being dominant due to the small number of data for this region in this month (not offset table). The SIF derived by Correction 1 was higher than that by the other corrections, and the SIFs derived by Corrections 3 and 4 agreed with the OCO-2 SIF, which was notably distinct from the trend observed for cropland, where Corrections 1 and 2 offered better agreement. There is a possibility that the

zero-level offset varies with land cover type even for regions in close proximity.

In this subsection, on the whole, GOSAT SIF agreed with OCO-2 SIF not only with respect to the seasonal cycle but also in terms of the absolute value for regions with high SIF emission. However, the difference in deviation from OCO-2 SIF between GOSAT offset correction methods varied over the sites. In addition, a large difference between GOSAT SIF and OCO-2 SIF was observed over several months and was not limited to November and December as it was in Case 1, even

when the number of GOSAT data were comparable with other months. In the next section we perform a comparison on a global scale to address these issues and to investigate whether the offset correction worked well in different locations around the world.

**4.4 Case 3: Comparison at global scale**

Figures 9 and 10 show the difference in the monthly mean SIF between GOSAT and OCO-2 within a $5° \times 10°$ grid box for

July and December 2015, respectively. In December, the difference between the correction methods was small. In July, the GOSAT SIF derived by Correction 1 was higher than the OCO-2 SIF, even for the latitude zone where the observation time was similar between the satellites. The difference in SIF became small when the zero-level offset correction was conducted separately for latitude zones (Correction 2) or conducted using bare soil data (Corrections 3 and 4). The observation time of GOSAT was later than that of OCO-2 in the northern high latitude region and earlier than that of OCO-2 in the southern

hemisphere. For Corrections 1 and 2, the GOSAT SIF was lower than the OCO-2 SIF in the northern high latitude region and higher than the OCO-2 SIF in the southern hemisphere. This may indicate that the different observation times between GOSAT and OCO-2 captured the diurnal cycle of the SIF yield from terrestrial vegetation. Generally, SIF varies mainly according to the absorbed PAR (APAR); however, this pattern changes under any environmental stress. One typical example is plants under water stress: increase in fluorescence emission stops in the early morning at a specific level of APAR, and

then the emission decreases toward late afternoon independent of APAR (Cerovic et al., 1996; Ounis et al., 2001; Flexas et al., 2002; Evain et al., 2004). In the present study, the difference in incoming PAR was accounted for by dividing SIF by cos(SZA). FAPAR (= APAR/PAR) was expected to be almost stable or to increase slightly with an increase in SZA during the time period including the observation time of GOSAT and OCO-2 (Kobayashi et al., 2012; Widlowski, 2010), which is





the opposite of the pattern of decrease in SIF yield for plants under water stress. The effect of observation geometry seems to be small, as discussed above. Therefore, the latitudinal pattern shown in Fig. 9 (a) and (b) is obtained if GOSAT and OCO-2 observe the terrestrial vegetation while the SIF yield is decreasing.

However, as shown in Fig. 11, this hypothesis does not always hold. For Correction 1 in both July and December,
GOSAT SIF was higher than OCO-2 SIF at a difference in observation time of around -1.5 h (southern hemisphere), including the grids where OCO-2 SIF was almost zero. The aforementioned hypothesis does not explain the difference in SIF for areas with almost no SIF emission. Thus, the higher GOSAT SIF seems to be caused by errors in the zero-level offset correction. According to Figs. 9 and 10, the separate correction for latitude zones (Correction 2) offered results similar to those by Correction 1. When the zero-level offset value derived from bare soil data was used (Correction 3), the zero level in
SIF agreed between GOSAT and OCO-2 (Fig. 11). There is a possibility that the zero-level offset differs among land cover types or regions, and bare soil is suitable to the correction for land areas.

For the northern high latitude region in July, GOSAT SIF was lower than OCO-2 SIF regardless of the correction methods (Fig. 9). Figure 11(a) shows that such lower GOSAT SIF was observed for the grids where there was a certain level of SIF emission, unlike in the aforementioned southern hemisphere. Furthermore, the tendency varied over the study years:
the difference between GOSAT SIF and OCO-2 SIF was significant in 2015. This seems to correspond to the photosynthetic activity of vegetation in this region: the satellite-based GPP (Luo et al., 2018) and the Soil Moisture Active Passive (SMAP) L4 carbon product (Madani et al., 2017) indicate the possibility of lower photosynthetic activity in 2015 compared to 2016. Although the influence of the zero-level offset correction might play some role in this effect (e.g., the different tendency between the cropland and the grassland in USA found in Case 2), it is inferred that the diurnal cycle of SIF yield was
reflected in the difference between GOSAT SIF and OCO-2 SIF for the northern high latitude region. This is supported by the results that lower GOSAT SIF compared to OCO-2 SIF was observed for Corrections 1 and 2 that seem to arise positive bias, and the difference in SIF varied over the study years.

The difference in the monthly mean SIF between GOSAT and OCO-2 within a $5° \times 10°$ grid box was averaged over the 20° latitude bin (Fig. 12). The difference in SIF showed a seasonal cycle when the zero-level offset was evaluated from
vegetation-free targets in both land and ocean and applied over the globe (Correction 1). The zero-level offset had a seasonal cycle with a maximum around June for the northern hemisphere (the opposite of the cycle for the southern hemisphere) as described in Sect. 3.4. The seasonal cycle was not removed if the mean zero-level offset value over the globe was used. A latitudinal gradient in the zero-level offset value (larger offset value for the lower latitude region) also remained. Correcting the offset separately for latitude zones (Correction 2) improved the results. However, the amplitude of variation appears to be
larger compared to Corrections 3 and 4, indicating the possibility that the cloudy ocean filling-in insufficiently accounts for the zero-level offset for land. When the zero-level offset value derived from bare soil data was used (Corrections 3 and 4), the difference varied around 0 with almost no clear seasonal cycle, although the difference in the observation time in the northern high latitude region and the southern hemisphere should be accounted for. The mean difference was within 0.1 mW $m^{-2}$ $nm^{-1}$ $sr^{-1}$ for most months and the variation between grids was about 0.2 mW $m^{-2}$ $nm^{-1}$ $sr^{-1}$. It should be noted that even





for Corrections 3 and 4, a somewhat large difference occurred in the latitude region where the difference in observation time was small, and a slight seasonal cycle remained for 40°N to 60°N.

The finding that the difference in SIF was close to 0 indicates agreement between GOSAT SIF and OCO-2 SIF for the latitude zone of 0° to 40°N where the difference in the observation time was small. This is also applicable to the southern hemisphere, as the GOSAT SIF that had a zero level similar to that of OCO-2 SIF (Corrections 3 and 4) did not show positive deviation. Most areas in the southern hemisphere, where the difference in the observation time was large, mainly consisted of barren areas with low SIF emission. In the northern high latitude region, the difference in the observation time and the characteristics of the vegetation seemed to result in a lower GOSAT SIF than OCO-2 SIF, although errors in the zero-level offset might be included.

Interpreting the time-averaged results (Fig. 13) from the above point of view, the GOSAT SIF that was the most consistent with OCO-2 SIF was obtained when the zero-level offset was evaluated from bare soil over the globe (Correction 3). The averaged difference was within 0.05 mW m$^{-2}$ nm$^{-1}$ sr$^{-1}$ and the month-to-month variation was about 0.05 mW m$^{-2}$ nm$^{-1}$ sr$^{-1}$. Concerning Correction 4, the narrow coverage of the TANSO-FTS radiance level and the small number of data for the offset table yielded a larger mean difference and standard deviation compared to Correction 3. Although Correction 3 seems to be suitable for deriving SIF, its coverage of the TANSO-FTS radiance level is limited. Therefore, latitudinal investigation of the filling-in signal over cloudy ocean (Correction 2) is a possible option to support the retrieval of gas concentration. More specifically, the apparent zero-level offset value that is retrieved simultaneously with the gas concentration (Frankenberg et al., 2012) can be compared with the zero-level offset of Correction 2. Even when Correction 3 was used, positive deviation from the OCO-2 SIF occurred in the latitude region with a similar observation time, and a seasonal cycle remained in the difference from the OCO-2 SIF in the northern high latitude region. In addition, the difference between GOSAT SIF Correction 3 and OCO-2 SIF varied between the cropland and the grassland in the USA (Sect. 4.3). More detailed investigation of the distribution of the filling-in signal according to land cover types and regions, including the difference between land and ocean, will be needed in the future.

## 5 Conclusion

The difference in the filling-in signal among data identified by various criteria was analysed for the zero-level offset correction to derive SIF from the GOSAT TANSO-FTS spectra. For cloudy ocean, data with a small variation in the TANSO-CAI radiance within the TANSO-FTS IFOV are suitable. For bare soil, data should be identified by the criteria based on the albedo of the TANSO-FTS bands 2 and 3 and the NDVI value. Different zero-level offset corrections (candidates for entirely vegetation-free area versus only bare soil areas and over the globe versus separately by latitude zone) were conducted, and the derived SIF was compared with OCO-2 SIF. GOSAT SIF generally agrees with OCO-2 SIF over areas with various SIF emissions regardless of the correction method, according to the comparison at the footprint level and in the specific vegetated regions with similar observation times. According to the investigation of the difference between



GOSAT SIF and OCO-2 SIF within a 5° × 10° grid box over the globe, a significant difference between the SIFs can be caused by a difference in the zero-level offset between land and ocean, even when the zero-level offset correction is conducted separately for latitude zones. The GOSAT SIF that was the most consistent with OCO-2 SIF was obtained when the zero-level offset was evaluated from bare soil over the globe. In this case, the difference between GOSAT SIF and OCO-2 SIF was within 0.1 mW m$^{-2}$ nm$^{-1}$ sr$^{-1}$ for most months with a variation among grids of about 0.2 mW m$^{-2}$ nm$^{-1}$ sr$^{-1}$. The difference was 0.05 mW m$^{-2}$ nm$^{-1}$ sr$^{-1}$ for the temporal average. Although a seasonal cycle was found in the zero-level offset evaluated from bare soil over the globe, the instrumental status is reflected in the variation of the zero-level offset, indicating a need for revaluation of the radiometric degradation in the TANSO-FTS after switching the optics path selector (Fig. 14).

Previous studies reported that the SIFs were consistent between GOME-2 and OCO-2, between TROPOMI and OCO-2, and between TanSat and OCO-2, while considering influential factors such as the observation time, location of the footprint, observation geometry, and wavelength. The comparison for GOME-2 was limited to the Amazon region, and the comparisons for TROPOMI and TanSat were initial-phase comparisons (Köhler et al., 2018b; Köhler et al., 2018a; Du et al., 2018). The present study compared GOSAT SIF and OCO-2 SIF under conditions in which the wavelength was similar between the sensors, cloudy data were excluded using the TANSO-CAI data for GOSAT, the land cover type and LAI within the TANSO-FTS footprint were taken into account. The comparison was conducted at multiple spatial scales to take into account the difference in the observation pattern between the satellites. The GOSAT SIF agreed with OCO-2 SIF with a bias of about 0.1 mW m$^{-2}$ nm$^{-1}$ sr$^{-1}$, which was comparable to the level of agreement in the aforementioned previous studies. Investigating the criteria for identifying vegetation-free areas was important to derive this small bias. Our results support the consistency among the present satellite-derived SIF data. It is also important that the consistency was confirmed between the FTS-derived and the grating spectrometer-derived SIF. Further studies will be needed to address the remaining bias, which seems to consist of instrumental effects, retrieval error, and characteristics of vegetation.

**Author contribution**

TM and YY conceived the framework of this research. YY carried out the retrieval of the filling-in signal from GOSAT spectra. HO conducted the data analysis and prepared the manuscript with contributions from all co-authors.

**Competing interests**

The authors declare that they have no conflict of interest.




## Acknowledgements

Retrieval of the filling-in signal was conducted at the GOSAT-2 Research Computation Facility. We would like to express our gratitude to NASA for making the MODIS and OCO-2 data available.

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





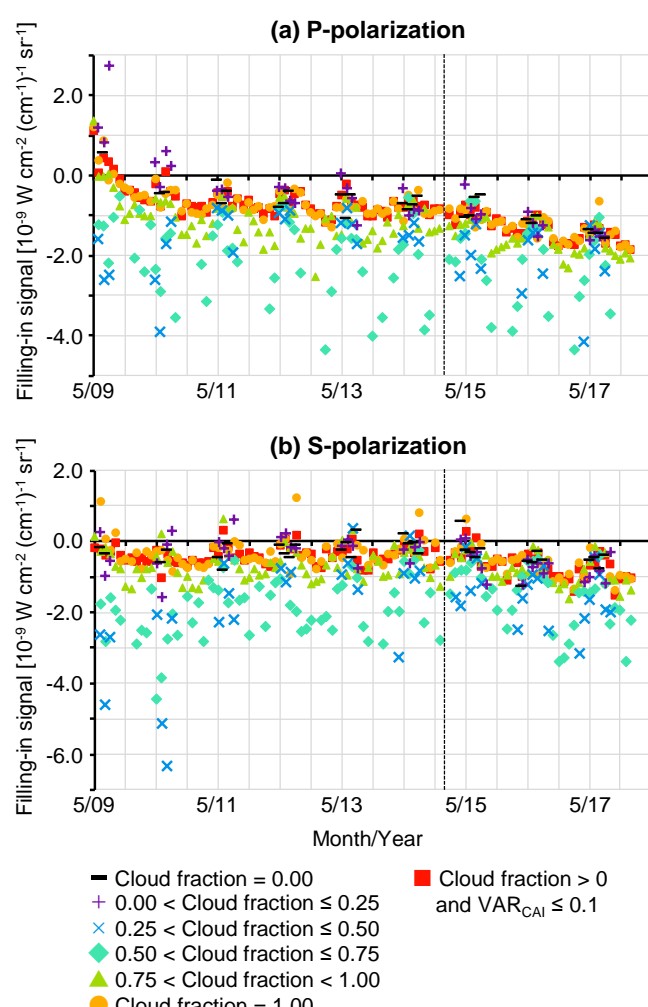

**Figure 1: Variation of the monthly mean filling-in signal derived from the ocean data having different cloud fractions within the latitude zone of 30°N to 45°N; (a) averaged radiance of the TANSO-FTS band 1 P-polarization 0.0005 to 0.0006 V (cm⁻¹)⁻¹; (b) S-polarization 0.0006 to 0.0007 V (cm⁻¹)⁻¹. A vertical broken line indicates the change of the optics path selector (26 January 2015).**





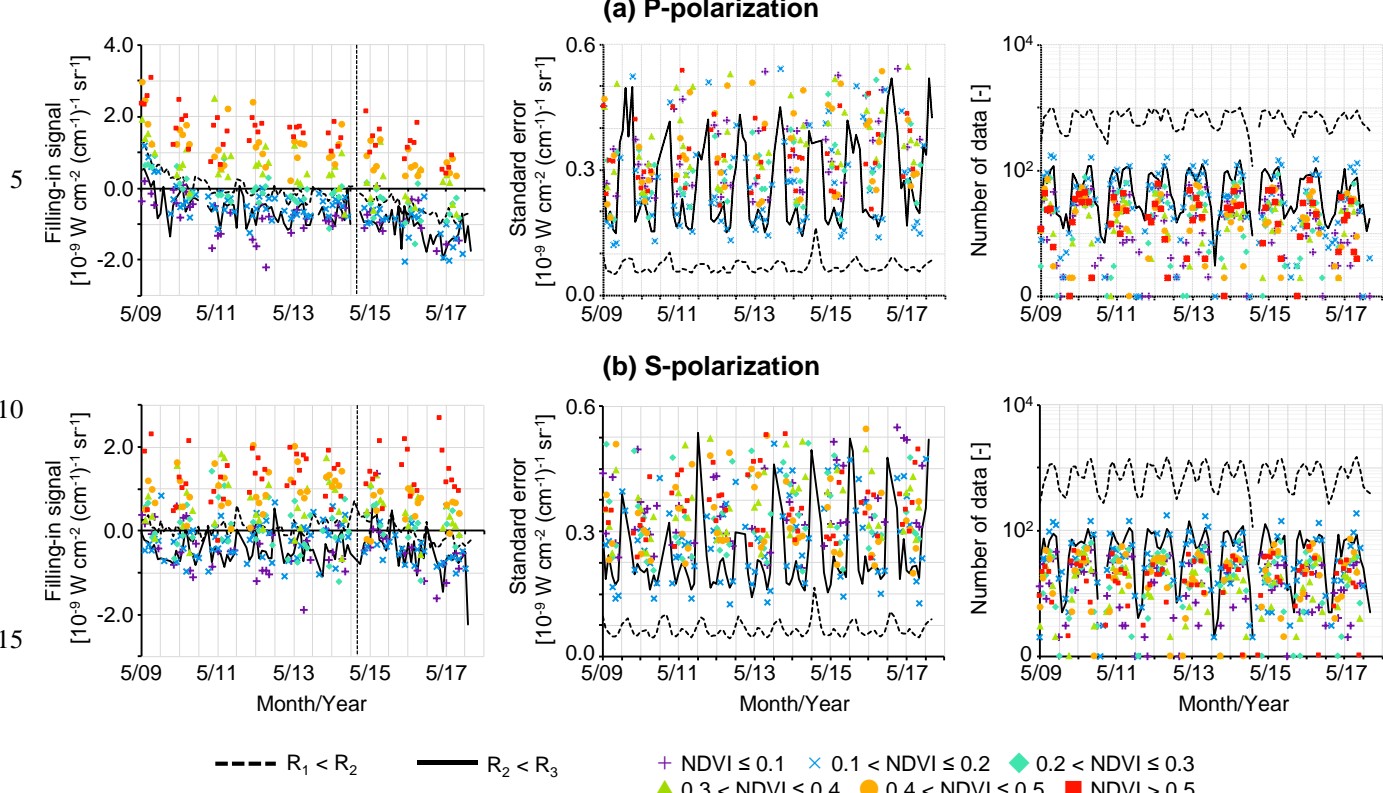

**Figure 2: Variation of the monthly mean filling-in signal derived from data identified by the different criteria, albedo values ($R_i$ for the TANSO-FTS band $i$) or NDVI values, within the latitude zone of 30°N to 45°N; (a) averaged radiance of the TANSO-FTS band 1 P-polarization 0.0005 to 0.0006 V (cm$^{-1}$)$^{-1}$; (b) S-polarization 0.0006 to 0.0007 V (cm$^{-1}$)$^{-1}$. A vertical broken line indicates the change of the optics path selector (26 January 2015).**

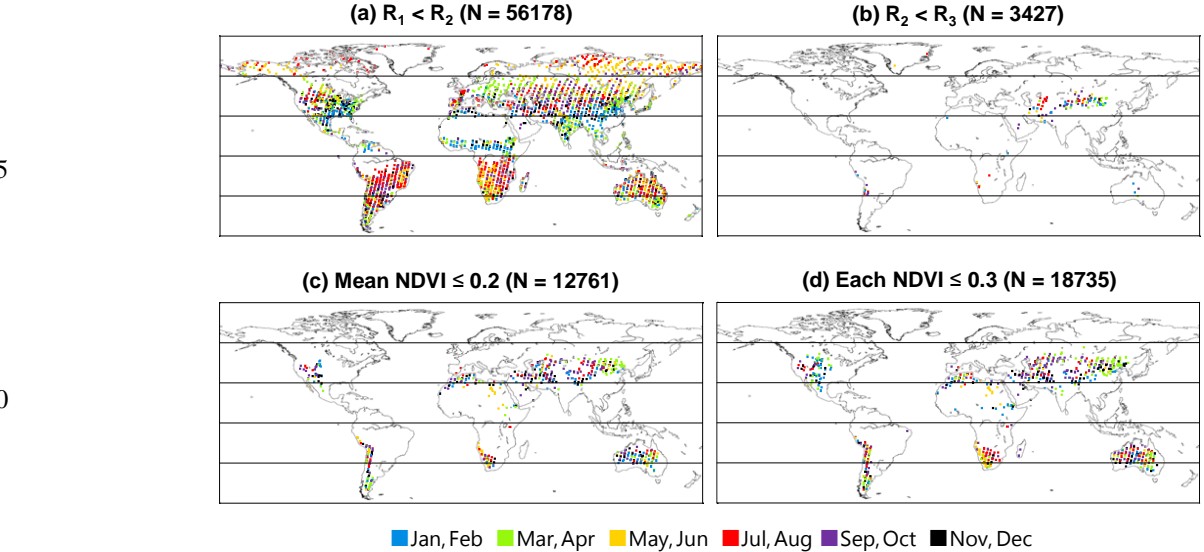

**Figure 3: Locations of data identified as the candidates for vegetation-free areas by the albedo values ($R_i$ for TANSO-FTS band $i$) or NDVI values for P-polarization in 2016; (a) $R_1 < R_2$; (b) $R_2 < R_3$; (c) mean NDVI ≤ 0.2; (d) NDVI$_{max}$ ≤ 0.3. The numbers of data points ($N$) are also presented.**

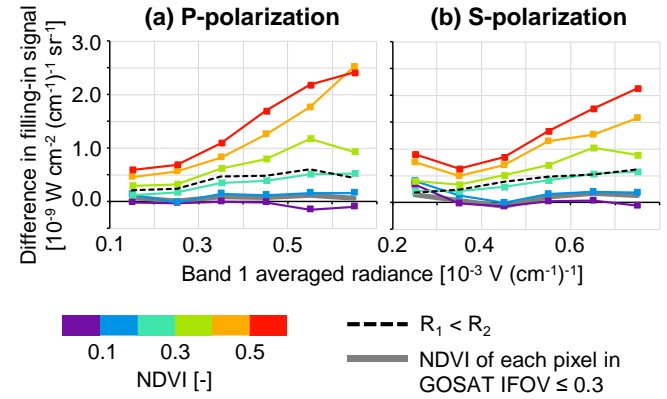

**Figure 4: Temporal mean (February 2015 to January 2018) of the difference in the monthly mean filling-in signal between data having different NDVI values and data with $R_2 < R_3$ ($R_i$: albedo for the TANSO-FTS band $i$), plotted against the averaged radiance of the TANSO-FTS band 1 for 30° N to 45° N; (a) P-polarization; (b) S-polarization. Results derived from the data with $R_1 < R_2$ (dotted black line) and data with NDVI$_{max}$ ≤ 0.3 (solid black line) are also plotted.**





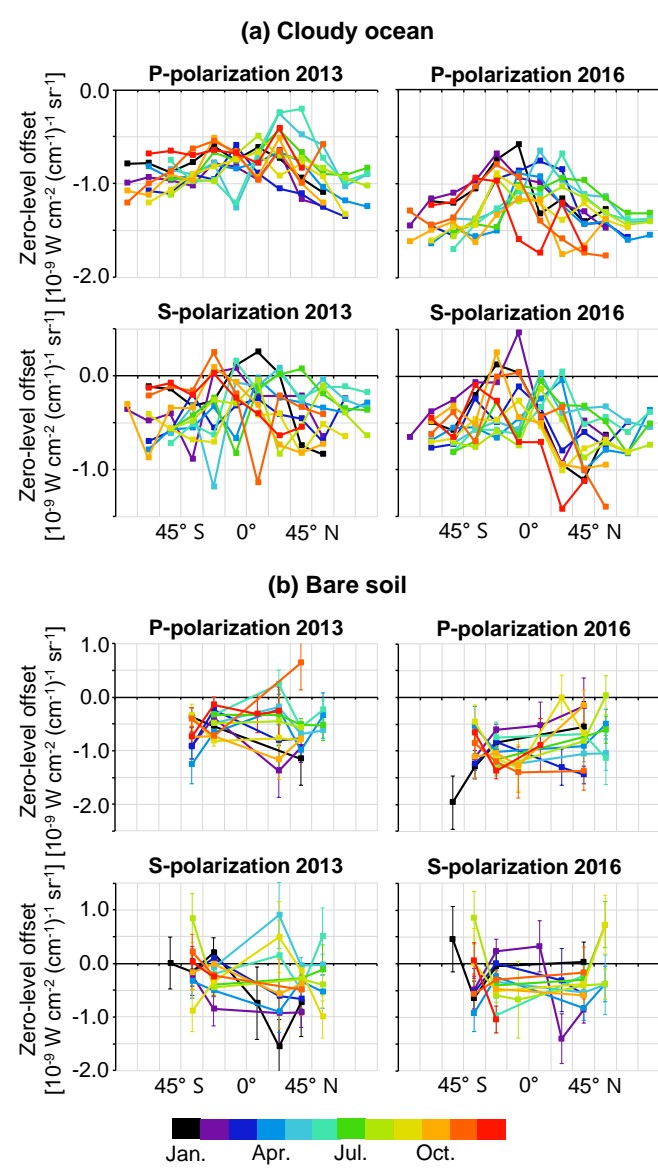

**Figure 5: Latitudinal variation of the monthly mean zero-level offset evaluated from (a) cloudy ocean data and (b) bare soil data. In each panel, results are presented separately for the averaged radiance of the TANSO-FTS band 1 P-polarization 0.0005 to 0.0006 V (cm⁻¹)⁻¹ and S-polarization 0.0006 to 0.0007 V (cm⁻¹)⁻¹ in 2013 and 2016.**





**Figure 6: Scatter plot of GOSAT SIF (Correction 1) and OCO-2 SIF whose footprints were included in the GOSAT TANSO-FTS footprint.**

**Table 1: Statistics of the relationship between GOSAT SIF and OCO-2 SIF. Slope and intercept are those of the linear regression lines. Mean difference (GOSAT SIF − OCO-2 SIF) and standard deviation of the difference are also shown.**

| | Number of data | Slope | Intercept | Correlation coefficient | Mean diff. [mW m⁻² nm⁻¹ sr⁻¹] | Std. of diff. [mW m⁻² nm⁻¹ sr⁻¹] |
|---|---|---|---|---|---|---|
| Correction 1 | 192 | 0.45 | 0.07 | 0.24 | -0.02 | 0.83 |
| Correction 2 | 191 | 0.48 | 0.06 | 0.25 | 0.01 | 0.85 |
| Correction 3 | 189 | 0.46 | 0.01 | 0.23 | -0.05 | 0.85 |
| Correction 4 | 177 | 0.48 | 0.03 | 0.25 | -0.02 | 0.83 |





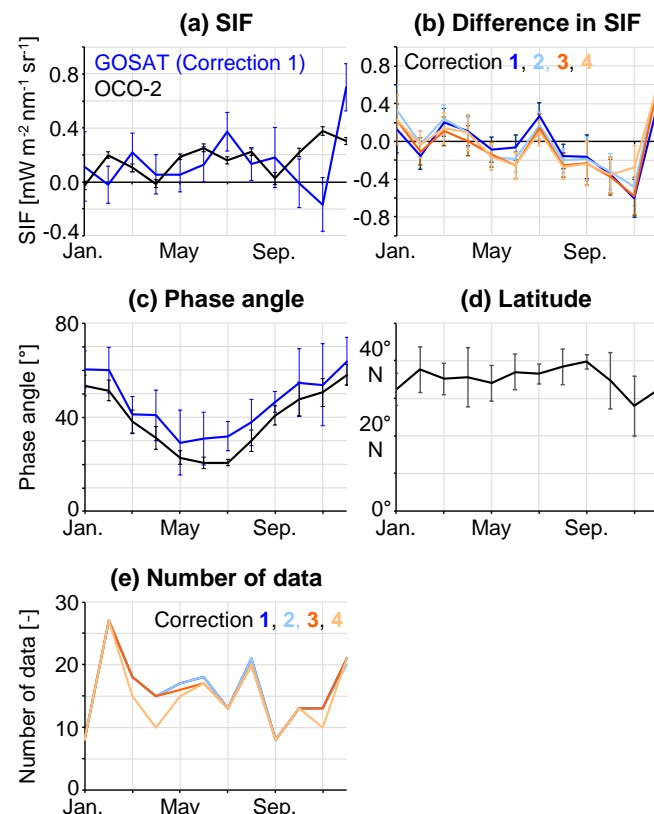

**Figure 7: Comparison between the GOSAT SIF and OCO-2 SIF whose footprints were included in the GOSAT TANSO-FTS footprint. Monthly mean values are presented for (a) SIF, (b) difference in SIF (GOSAT − OCO-2), (c) phase angle (angle between the observation direction and the incident sunlight), (d) latitude of the centre of the GOSAT TANSO-FTS footprint, and (e) the number of data points. For panels (b) and (e), results from each correction method are separated by colour. The error bar represents the standard error and the standard deviation for panels (a) and (b) and panels (c) and (d), respectively.**





**(a) Southeast Asia evergreen broadleaf**

**(b) USA cropland**

**(c) USA grassland**

GOSAT ▬▬ Correction 1 ▬▬ Correction 2 ▬▬ Correction 3 ▬▬ Correction 4  OCO-2 ▬▬ Nadir ▬ ▬ ▬ Glint





**Figure 8: Comparison of the monthly mean SIF between GOSAT and OCO-2 for specific regions where the difference in the observation time is small; (a) evergreen broadleaf forest in Southeast Asia; (b) cropland in USA; (c) grassland in USA. In each panel, the target region, SIF value, standard error (SE), phase angle (angle between observation direction and incident sunlight), LAI, and difference in local time of the observation (GOSAT − OCO-2) are presented. Only months with more than 4 data points are plotted.**



**Figure 9: Difference in the monthly mean SIF (GOSAT − OCO-2) within a 5° × 10° grid box for July 2015; (a)–(d) GOSAT SIF Correction 1–4 respectively; (e) difference in local time of the observation (GOSAT − OCO-2). Only grids with more than 9 data points are depicted.**





**(a) Correction 1**

**(b) Correction 2**

**(c) Correction 3**

**(d) Correction 4**

-0.7   0   0.7
Difference in SIF
(GOSAT−OCO-2)
[mW m⁻² nm⁻¹ sr⁻¹]

**(e) Difference in observation time**

-2   0   2
Difference in observation time
(GOSAT−OCO-2) [h]

**Figure 10: Similar to Fig. 9 but for December 2015.**



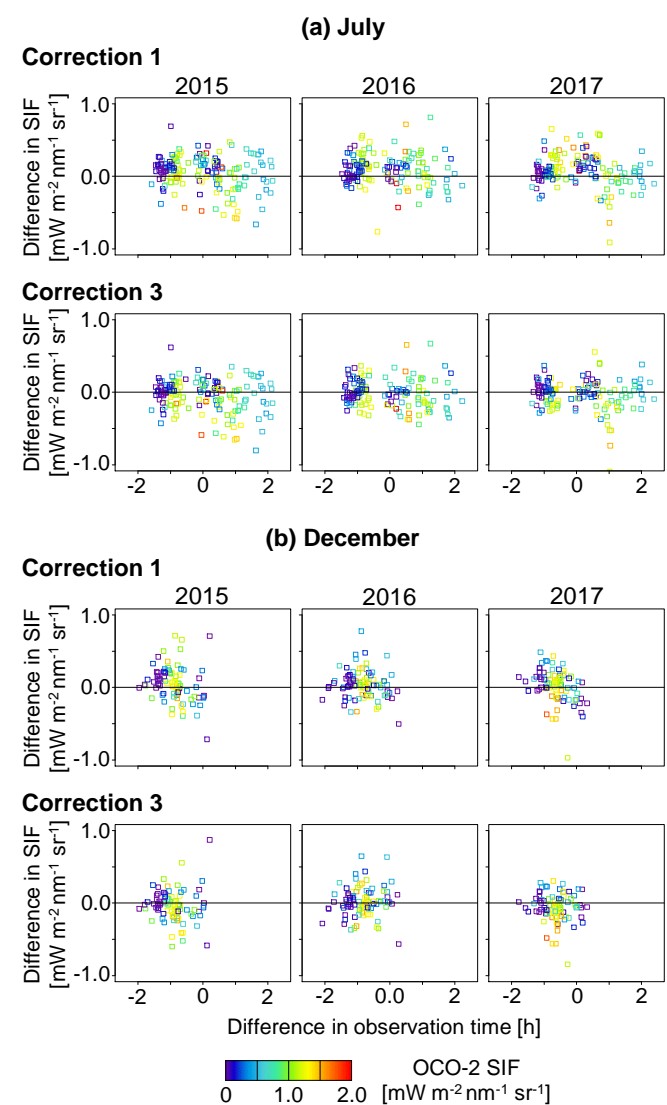

**Figure 11: Relationship between the difference in SIF (GOSAT − OCO-2) and the difference in local time of the observation for 2015 to 2017; (a) July; (b) December. In each panel, results from the GOSAT SIF Corrections 1 and 3 are presented. Each symbol represents a 5° × 10° grid box.**





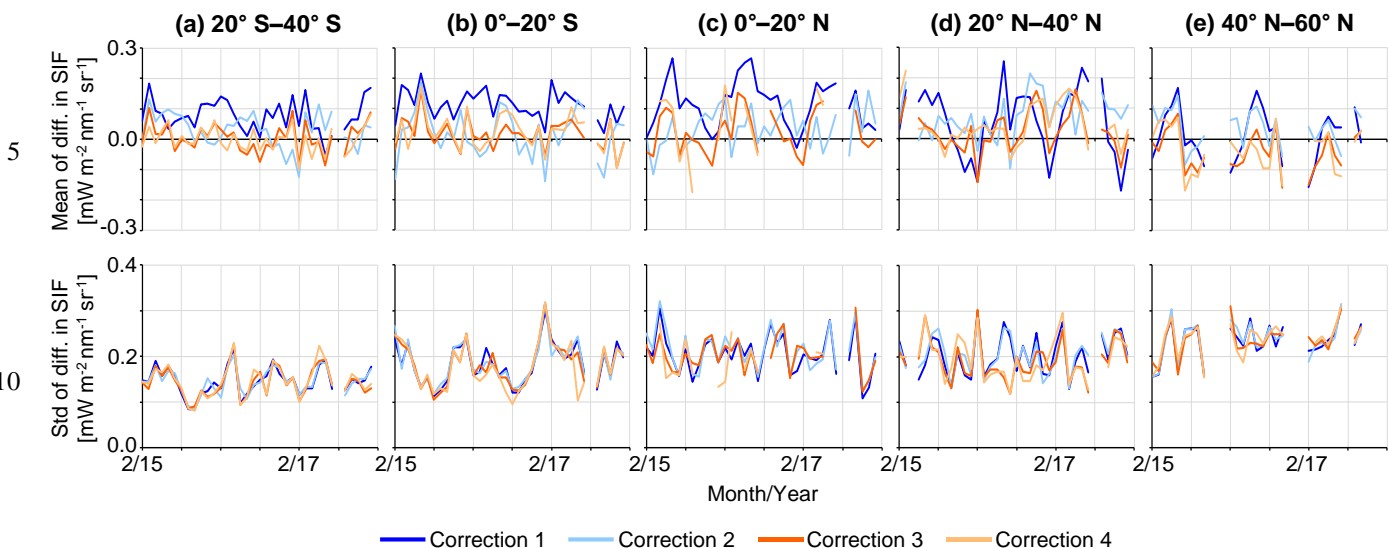

**Figure 12: Difference in the monthly mean SIF (GOSAT − OCO-2) within a 5° × 10° grid box is averaged for each 20° latitude bin; (a)–(e) 20°S–60°N with an interval of 20°. In each panel, the averaged value and the standard deviation of the difference in each bin are presented.**





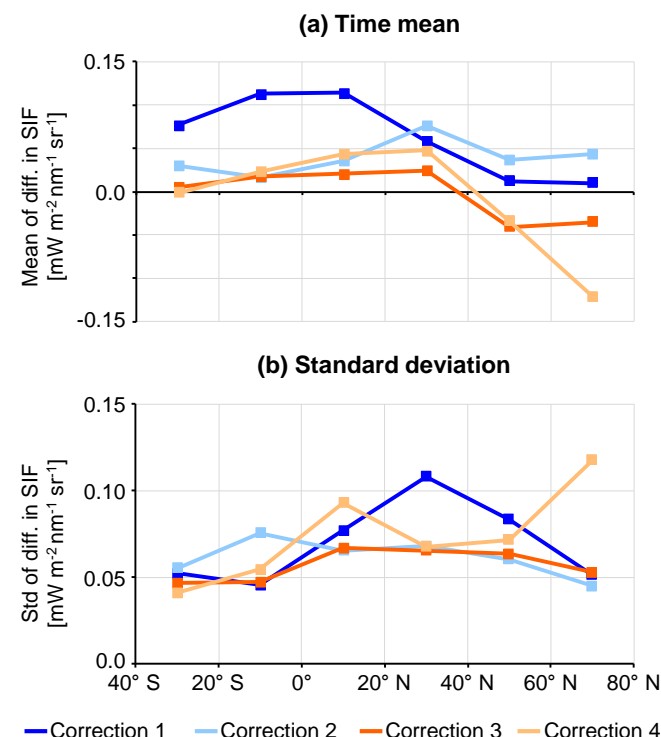

**Figure 13: The monthly mean value shown in Fig. 12 is averaged over the target time period (February 2015 to January 2018); (a) averaged value; (b) standard deviation.**





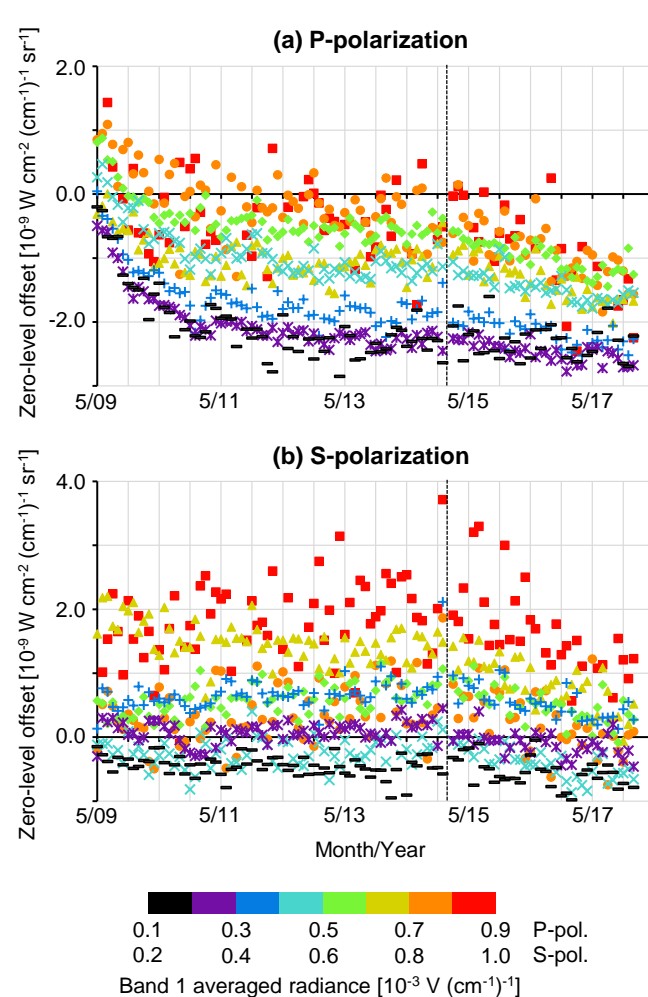

**Figure 14: Variation of the monthly mean zero-level offset (filling-in signal evaluated from bare soil over the globe) calculated against bins of the averaged radiance of the TANSO-FTS band 1; (a) P-polarization; (b) S-polarization. A vertical broken line indicates the change of the optics path selector (26 January 2015).**

