# Peer review of "On the zero-level offset in the GOSAT TANSO-FTS O2 A-band and the quality of solar-induced chlorophyll fluorescence (SIF): Comparison of SIF between GOSAT and OCO-2"

_Atmospheric Measurement Techniques, 2019_

## Referee Comment (RC1) · Luis Guanter (Referee) · 3 Sep 2019

The study by Oshio et al. analyzes the impact of the so-called zero-level offset (ZLO) on SIF retrievals from GOSAT TANSO-FTS near-infrared spectra. The ZLO is an additive signal with a similar in-filling effect of near-infrared absorption lines as SIF has, which causes the ZLO can bias SIF retrievals. The first part of the manuscript deals with the characterization of the ZLO from data acquired over different non-fluorescent targets (either the cloudy ocean or bare soils), whereas the second part compares SIF retrievals from GOSAT under different ZLO corrections with OCO-2 SIF data at different

spatial scales.

I can't identify any methodological flaw in this study, the text is well written and the results are clearly presented and discussed. My only major concern is on its very narrow scope and lack of highly significant findings: it deals with a GOSAT-specific instrumental issue (the ZLO) for which several characterization and correction methods have already been published, SIF data from GOSAT are no longer so widely used after the advent of GOME-2, OCO-2 and TROPOMI, and in any case the results of the analysis seem to show that the accuracy of the GOSAT SIF data is relatively robust against which ZLO characterisation/correction method is used in the processing. The most interesting result in my opinion is the very good correspondence between GOSAT and OCO-2 SIF data, which suggests the user community could easily combine both data sets to produce longer time series of "high spatial resolution" SIF data (as opposed to the "low resolution" of GOME/GOME-2/SCIAMACHY).

Here there are some comments that the authors might like to consider in their revision of the manuscript (at their discretion):

- Format: The manuscript could be reformatted as a shorter technical note simply presenting the most meaningful ZLO characterization approach among those tested (the one over bare soils for different latitudinal belts, I think) and the subsequent good comparison between the GOSAT and OCO-2 SIF products when that ZLO characterization is used. I believe that several sub-sections and figures could be moved to the Supporting Information without harming the rigor and readability of the manuscript. For example, whether cloudy skies or barren areas are better to characterize ZLOs for latter SIF retrievals is a very specific research question, and the paragraph "To date ... OCO-2 SIF (Köhler et al, 2018a)" could be removed from the Introduction.

- Introduction: In p.2 L14 it is mentioned that ZLOs in GOSAT are actually not only important for SIF retrieval, but also for XCO2 and XCH4 retrievals (core GOSAT products). Why is then this study solely focused on SIF? Extending it to XCO2 and XCH4 would

actually solve my "narrow scope" concern mentioned earlier, although I am aware that it would then require to fully rewrite the manuscript. It might also be good to mention GOSAT-2 in the introduction, and adding some preliminary assessment of ZLOs in GOSAT-2 would also help enhance the impact of the manuscript.

- Comparision of TOA radiances from GOSAT and OCO-2: it would be interesting to see how near-infrared TOA radiances from TANSO-FTS and OCO-2 compare. A radiometric offset in TOA radiance should translate into the same offset in SIF. Do the near-infrared radiances from GOSAT and OCO-2 compare within 0.1 mW/m2/sr/nm as the SIF products do? Is there a bias between the radiances from the two instruments? If so, how is that translating into biases in SIF? A comparison of TOA radiances could also help assess the effect of varying illumination and observation geometries between the two systems.

- p.5, L14: How exactly were the different view zenith angles in GOSAT (up to $\sim30°$?) taken into account in the comparisons with nadir OCO-2 data? It is mentioned that "It is difficult to investigate the influence of observation geometry on SIF..." in p.11 L8.

- p.8, L15: Guanter et al. (2012) reported on an apparently clear decrease of ZLOs with time for both S and P polarizations at 755 nm (not at 770 nm), which seemed to compare well with the temporal evolution of the spectral slope of the radiance spectrum at the same wavelengths. Could the authors confront those findings with their own results, perhaps expanding their analysis to also include a test on spectral slopes (could be included in the Supporting Information)?

---

## Referee Comment (RC2) · Christian Frankenberg (Referee) · 11 Oct 2019

**Christian Frankenberg (Referee)**

cfranken@caltech.edu

Received and published: 11 October 2019

The study by Oshio et al presents an in-depths analysis of instrumental effects of the GOSAT FTS instrument and its effects on SIF retrievals from space. The 0-level offset effect in FTS system has been previously found to have a major impact on SIF retrievals (as well as trace gas retrievals in general for that matter), so the topic is warranted and suitable for AMT. In general, I find the paper to be well written and comprehensive and I recommend publication in AMT after some minor (some may be a bit more involved) concerns are resolved. Even though there are no really new scientific results, the

technical nature of the submission and the journal should alleviate this concern.

Big picture concerns:

In general, I see the problem mostly as a problem of noise, which makes a conclusive statement often impossible. The different choices for reference regions yield dramatically different numbers of retrievals, resulting often in rather high standard errors (due to the low data rate from GOSAT). In many case, differences are thus not significant and there are a number of plots for which I don't really see the point. This holds for Fig5: What do I really see here? It all seems noisy and it is unclear whether (and why) any specific patterns are reliable. Fig 6: This looks like a shot-gun blast, from which I don't learn anything, Can you co-add by OCO-2 SIF bin and then plot aggregated data? Single soundings are often meaningless unless averaging is applied. Fig 7 (Differences between OCO-2 and GOSAT seem to be all within +/- 2 sigma, so no conclusions can be drawn). Figs 11/12: Again, not really too clear patterns here. I think some of the plots can be omitted and the story streamlined to the major aspects. The discussion should include whether any patterns are statistically significant (or not)

Given that the problem in the right 0-level offset correction is that it is noisy, a primary goal would be to have as low of a standard error in the 0-level offset per radiance bin and temporal bin. The authors could put some more effort into discussing this aspect and which analyses might help (e.g. temporal and radiance bin smoothing, etc). Eventually, the differences they see for the various procedures might be mostly within the noise

Specific comments:

P3, Line 28: Is the Low gain actually used? Also worthwhile to mention that OCO-2 can use the Sahara as a reference while GOSAT can't, as the gain changes. Again, this increases noise in the correction for GOSAT.

P4/L8: Why was 771nm not retrieved? It would reduce noise in final SIF data
P5/L6: Please add Sun et al Science publication here

P6/L24: Why did you choose these radiance bins exactly? (vegetation is often brighter). Is there a difference in mean radiance for all methods in Figure 1 (you have the selected min/max selection but you might not sample the same radiance averages, would be good to look at that as well. Also, we found that this radiance bin is less time dependent as lower radiance.

P9/L28: Is the SI figure plotted at a bin of 0.00003V/cm-1? If yes, it seems to coarse as some of the features we found in GOSAT data are highly dependent on radiance level and might warrant finer sampling. In your SI figure, it seems too coarse.

---

## Author Comment (AC1) · 12 Nov 2019

Please see the supplement for the response and revision.

Please also note the supplement to this comment:
https://www.atmos-meas-tech-discuss.net/amt-2019-234/amt-2019-234-AC1-supplement.zip

---

## Author Comment (AC2) · 12 Nov 2019

Please see the supplement for the response and revision.

Please also note the supplement to this comment:
https://www.atmos-meas-tech-discuss.net/amt-2019-234/amt-2019-234-AC2-supplement.zip